# The Enterochromaffin-like [ECL] Cell—Central in Gastric Physiology and Pathology

**DOI:** 10.3390/ijms20102444

**Published:** 2019-05-17

**Authors:** Helge L. Waldum, Øystein F. Sørdal, Patricia G. Mjønes

**Affiliations:** 1 Department of Clinical and Molecular Medicine, Faculty of Medicine and Health Sciences, Norwegian University of Science and Technology, N-7491 Trondheim, Norway; 2 Department of Gastroenterology and Hepatology, St. Olav´s Hospital-Trondheim University Hospital, N-7491 Trondheim, Norway; OSORDAL@rcsi.ie; 3 Department of Pathology, St. Olav´s Hospital- Trondheim University Hospital, N-7491 Trondheim, Norway; patricia.mjones@ntnu.no

**Keywords:** ECL cell, gastric acid, gastrin, neoplasia

## Abstract

Background: Studies on the regulation of gastric and pancreatic secretion began more than 100 years ago. Secretin was the first hormone postulated to exist, initiating the field of endocrinology. Gastrin produced in the antral mucosa was the second postulated hormone, and together with histamine and acetylcholine, represent the three major gastric acid secretagogues known since 1920. For a long time, the mast cell was the only recognized histamine-producing cell in the oxyntic mucosa and, in the mid-1980s, the ECL cell was recognized as the cell producing histamine, taking part in the regulation of gastric acid secretion. Methods: This review is based upon literature research and personal knowledge. Results: The ECL cell carries the gastrin receptor, and gastrin regulates its function (histamine release) as well as proliferation. Long-term hypergastrinemia results in gastric neoplasia of variable malignancies, implying that gastric hypoacidity resulting in increased gastrin release will induce gastric neoplasia, including gastric cancer. Conclusions: The trophic effect of gastrin on the ECL cell has implications to the treatment with inhibitors of acid secretion.

## 1. The Regulation of Gastric Acid Secretion and the Cell in Oxyntic Mucosa Producing and Releasing the Histamine Taking Part in This Regulation: A Historical Overview.

In many ways modern physiology started with studies on the regulation of gastric acid secretion. Both patients and doctors became interested in gastric acid since many experienced symptoms easily related to gastric acidity, like heartburn and regurgitation of acidic gastric content. Moreover, gastric juice may be collected from humans and animals. Beaumont studied the regulation of gastric acid secretion in a person with a gastric fistula to the skin [1], and Pavlov started experimental animal physiology by studying gastric fistula dogs [2]. He showed the importance of the nerves (vagal nerves) and acetylcholine in the stimulation of gastric acid secretion. Shortly afterwards, Edkins postulated the presence in the antrum of a hormone stimulating acid secretion in the oxyntic mucosa and named it gastrin [3]. Finally, Popielsky described the third major gastric acid secretagogue, histamine, in 1920 [4]. The interaction between these three gastric acid secretagogues was disputed for decades. Kahlson showed that food intake as well as gastrin reduced the concentration of histamine in the oxyntic mucosa, and at the same time stimulated the synthesis of histamine [5]. The mast cell was the only recognized histamine-producing cell in the oxyntic mucosa and, thus, assumed to take part in the regulation of gastric acid secretion. However, the mast cell is a cell without any fixed position in relation to the acid producing parietal cell. In the late 1960s Håkanson et al. reported that a neuroendocrine (NE) cell in the oxyntic mucosa of rodents contained histamine, and they named this cell the enterochromaffin (EC)-like (ECL) cell due to the resemblance to the EC cell in the small intestine [6]. In spite of this description, the mast cell remained to be the putative histamine-producing cell in the oxyntic mucosa of humans until many years later. Then, Håkanson’s group, after having improved the sensitivity of histamine determination, reported that there was histamine also in ECL cells in humans [7]. At that time, it was already known that long-term profound acid inhibition induced by either proton pump inhibitors (PPI) [8] or an insurmountable histamine-2 receptor antagonist [9], induced malignant ECL cell-derived tumours in the oxyntic mucosa of rodents. Since these compounds are chemically unrelated, it was evident that the tumorigenic effect was due to their biological effect, inducing hypoacidity with secondary hypergastrinemia [10]. These findings obviously showed that the ECL cell in rodents had a gastrin receptor transmitting the positive trophic effect. Moreover, physiological studies demonstrated that gastrin released histamine from isolated oxyntic cells [11,12] as well as that the histamine release from isolated rat stomach correlated with the ECL cell density [13]. Nevertheless, many scientists with influence still claimed that the gastrin receptor was located on the parietal cell. This view got important support when the gastrin receptor was cloned from isolated oxyntic mucosal cells purified to 95% in parietal cells [14]. However, the cell fraction was not completely pure, and contamination with only one ECL cell could have been sufficient to affect the cloning result. In the isolated rat stomach, using a fluorescein-labelled gastrin analogue at a near-physiological concentration of 100 pmol/L, we could demonstrate binding to the ECL cell and not the parietal cell, finally showing that there was no gastrin receptor on the parietal cell [15]. Later, Athmann and co-workers from the group of Sachs, studying intracellular Ca^2+^ release in isolated rabbit oxyntic cells, reported that gastrin up to a concentration of 1000 pmol/L induced a Ca^2+^ increase which could be blocked by a histamine-2 blocker [16]. When increasing the gastrin concentration further to 10,000 pmol/L, they could not completely block the Ca^2+^ increase by a histamine-2 blocker, and therefore they expressed some ambiguity concerning a gastrin receptor on the parietal cell. However, this low affinity for gastrin is not compatible with a gastrin receptor but could be due to interaction with another receptor. It may, therefore, be concluded that gastrin stimulates gastric acid secretion by stimulating ECL cell histamine release, and that the gastrin receptor is localized to the ECL cell and not the parietal cell. The central role of gastrin in the regulation of gastric acid secretion has also been demonstrated in mice with a knocked-out gastrin gene [17] or gastrin/CCK-B receptor gene [18]. Neither gastrin nor histamine nor carbachol stimulated acid secretion in the gastrin knockout mice. However, gastrin injections for some days partly restored gastric acid secretion [17]. This finding could, nevertheless, indicate the presence of a gastrin receptor on the parietal cell. An alternative explanation is that the parietal cell function is dependent on histamine or Reg-1 protein [19] or another mediator from the ECL cell. The ECL cell has prolongations approaching the parietal cell in a synaptic-like manner [20]. It should also be added that the ECL cell produces calbindin [21], and at least upon gastrin stimulation, the alfa-glycoprotein hormone [22].

## 2. Embryology of the ECL Cell.

Pearse and Polak formulated the neural crest theory for the origin of NE cells [23]. However, LeDouarin and Teillet could not find evidence for neural crest origin of NE cells in the gastrointestinal tract when studying chimeric birds (chicken and quail) [24] and, therefore, NE cells in the gastrointestinal tract are presumed to be of endodermal origin. Nevertheless, NE cells have important differences compared with other mucosal cells since they can proliferate [25], and some may be pluripotent [26]. Very recently it was reported that the EC cell in the gut could show stem cell-like qualities [27], probably also of relevance to the ECL cell. Although the study by LeDouarin and Teillet [24] is important in excluding the neural crest origin of gastrointestinal NE cells, these cells are very similar to pituitary cells accepted as being of neural crest origin. It is still a possibility that neural (crest) cells could have migrated at an earlier stage and before the establishment of the chimeric birds. Nevertheless, the NE cells in the oxyntic mucosa, including the ECL cell, are different from the NE cells in the gut and antral mucosa, being independent of neurogenin-3 in their development [28]. In the rat, histidine decarboxylase (HDC)-positive cells can be recognized at embryonic day 16, successively followed by the appearance of histamine, chromogranin A, and vesicular monoamine transporter 2 [29]. Gastrin receptors have been detected by immunohistochemistry in foetal undifferentiated gastric epithelial cells, but not on the stem cells of adult rats [30]. ECL cells expressed this receptor, whereas gastrin receptor immunoreactivity on parietal cells was ambiguous [30]. In any way, the ECL cell can proliferate and develop into neoplasia, and gastrin has a positive trophic effect on this cell through its gastrin receptor. 

## 3. Functional and Trophic Regulation of the ECL Cell

The main regulator of the ECL cell function, that is histamine release, is gastrin. Gastrin released by food intake, has a high potency reflecting high affinity to the gastrin receptor. This is demonstrated by detectable stimulation of histamine release at a concentration of 2 pmol/L [31]. Moreover, maximal effect is reached at about 500 pmol/L, similar to the trophic effect in rats [32], and to the functional [33] and trophic [34] effects in humans. There is no threshold for the trophic effect of gastrin and, consequently, only moderate hypergastrinemia as usually seen during treatment with inhibitors of gastric acid secretion, induces ECL cell hyperplasia [35]. The increase in ECL cell mass results in increased histamine release [13], which is responsible for the rebound acid hypersecretion seen after stopping treatment with PPIs [36]. The cholinergic agent McN-A-323 (muscarinic-1 analogue) did not stimulate histamine release, whereas vagal stimulation increased histamine release in the isolated rat stomach, although not sufficiently completely explained the stimulation of acid secretion [37]. The vagal stimulation of histamine release is probably mediated by PACAP (pituitary adenylate cyclase-activating polypeptide) released from neurons [38]. Like gastrin, PACAP has a trophic effect on the ECL cell that could be responsible for the trophic effect of the vagal nerves, elegantly shown by unilateral vagotomy in rats by Håkanson and co-workers [39]. The role of PACAP in the trophic regulation of the ECL cell is, however, difficult to assess since, presently, there is no method to quantitate vagal activity. Somatostatin inhibits histamine release and, at the same time, parietal cell H+ secretion, making it a very efficient inhibitor of acid secretion [40]. Generally, there is a close relationship between functional and trophic regulation of a cell type, which is logical from a biological point of view. This is also the case for the ECL cell. Thus, gastrin not only stimulates histamine release, but also proliferation of the ECL cell, effects mediated by the same gastrin receptor and accordingly showing similar concentration dependence. Long-term hypergastrinemia leads to ECL cell hyperplasia, manifested by rebound acid hypersecretion [36] and in the long-term by ECL cell-derived tumours of variable malignancies [38]. Gastrin undoubtedly predisposes to ECL cell neuroendocrine tumours (NETs). There are also arguments for a role of gastrin in the pathogenesis of gastric carcinomas, as patients with autoimmune gastritis have increased risk of malignancy [41,42]. Moreover, patients with hypergastrinemia upon long-term follow-up have an increased prevalence of gastric cancer [43]. PACAP is probably responsible for the functional and trophic effects of the vagal nerves on the ECL cell [44]. Long-term treatment with the long-acting somatostatin analogue octreotide reduced ECL cell density in hypergastrinemic patients secondary to autoimmune gastritis [45]. Octreotide also reduced ECL cell density in hypergastrinemic rats due to dosing with a peroxisome proliferator type α, without affecting the gastrin concentration [46].

## 4. Role of the ECL Cell in Disease

### 4.1. Acid-Related Disorders

#### 4.1.1. Rebound Acid Hypersecretion.

Profound acid inhibition for some time induces rebound hypersecretion of acid after stopping treatment [36]. Due to the prolonged effect of PPIs, this rebound effect became detectable at first after two weeks without PPI. Therefore, PPIs were initially claimed not to induce rebound acid hypersecretion secretion since acid secretion was determined too early after stopping treatment. Healthy individuals develop dyspepsia in the period of rebound acid hypersecretion [47], and it is probable that patients with gastro-oesophageal reflux disease (GERD) will also have increased symptoms in this phase. The problem in stopping with PPIs when started [48] is most probably due to the acid rebound effect. The ECL cell hyperplasia also reduces the sensitivity for histamine-2 (H-2) receptor blockers [49], which is an important argument for starting with H-2 blockers in most of the patients with GERD. 

#### 4.1.2. Peptic Ulcer Disease

*Helicobacter pylori* (Hp) is the main cause of peptic ulcer disease [50]. The pathogenesis differs between duodenal and gastric ulcer. In duodenal ulcer disease, the acid secretion is increased due to Hp infection in the antrum leading to NH_3_ production and stimulation of gastrin release [51]. The slight hypergastrinemia stimulates acid secretion. Moreover, there is an inappropriate hypergastrinemia in relation to gastric acidity [52], probably caused by an increased ECL cell mass and histamine release due to the slight hypergastrinemia [53]. The fall in maximal (penta)gastrin stimulated acid secretion seen after Hp eradication in patients with duodenal ulcer probably reflects a fall in ECL cell density since maximal gastrin stimulated acid secretion is dependent on the ECL cell mass. On the other hand, the ECL cell is not directly involved in the pathogenesis of gastric ulcer besides taking part in the normal regulation of acid secretion. Hp predisposes to gastric ulcerations by reducing the defense mechanisms in the mucosa [mucus and bicarbonate secretions] [53].

#### 4.1.3. Gastroesophageal Reflux Disease (GERD)

Incompetence of the gastro-oesophageal valve function is the most important pathogenic factor for GERD. However, the acidity of the gastric content is also of utmost importance as evidenced by the efficiency of inhibitors of gastric acid secretion in the treatment. Moreover, increased gastric acidity per se may induce GERD as demonstrated in patients with gastrinoma [54], and indirectly by the subjects who developed dyspepsia during the rebound acid hyper-secretory phase [47]. The importance of keeping gastric pH above 4.0 [55] most likely reflects the important role of pepsin in the damage of the oesophageal epithelium since pepsin is destroyed above this pH.

### 4.2. Gastric Neoplasia

#### 4.2.1. Neuroendocrine Neoplasms (Neuroendocrine Tumours and Neuroendocrine Carcinomas)

The interest for gastric neuroendocrine neoplasms [NENs] started in the 1870s when endoscopic examination of the upper gastrointestinal tract became available. Gastric neuroendocrine tumours (NETs) are usually small and with a low malignant potential. Nevertheless, they can metastasize and kill the patient. Gastrin plays a central role in their pathogenesis since hypergastrinemia secondary to autoimmune gastritis with hypoacidity [34,56] and gastrinoma [57] with hyperacidity predispose to these tumours. Thus, gastric NETs are hormone-dependent tumours [58]. With the development of efficient inhibitors of gastric acid secretion like the PPIs [8] and insurmountable H-2 blockers [9], long-term dosed rodents developed NETs in the oxyntic mucosa, and the gastrin hypothesis was developed [10]. Practically all gastric NETs develop from the ECL cell, which in the normal state is the most prevalent NE cell in the oxyntic mucosa. To our knowledge, there is one gastric oxyntic NET described that originated from A-like cells, a ghrelinoma [59], and we have described a D-cell-derived tumour in a patient many years after resection of the antrum [60]. The prevalence of gastric NETs has increased during the last decades [61], possibly related to the increased use of efficient inhibitors of acid secretion. In early phases, PPI-induced gastric NETs are apparently reversible when stopping PPI treatment since they macroscopically disappear [62]. Similarly, at least some gastric NETs, due to atrophic oxyntic gastritis, vanish after reducing gastrin by antrectomy [63]. The most common cause of gastric NETs is autoimmune gastritis [34,56], but they are also reported to occur in patients with Hp gastritis [64,65]. As expected from the tumorigenic effect being due to a peptide hormone, the maximal tumorigenic effect is reached when all the receptors are loaded with their ligand. For the gastrin receptor, this is about 400 pmol/L [34]. The gastrin receptor antagonist netazepide [66], as well as long-acting somatostatin analogues [45], are efficient in the treatment of gastric NETs. Unfortunately, after stopping treatment NETs reappear due to unopposed hypergastrinemia [67]. Gastric NETs occur after long-term hypergastrinemia due to whatever cause in whatever species [68]. The relationship between gastric NETs and the much more malignant neuroendocrine carcinomas (NECs) is controversial, since the latter are found in patients without hypergastrinemia [69]. Nevertheless, we have described a patient with gastric NET secondary to autoimmune gastritis, developing into a highly malignant cancer [70]. In the Spanish family with both parents having a mutation in the same proton pump gene, the homozygote children developed gastric NETs and gastric NEC [71,72]. A relationship between an ordinary NET and a highly malignant one has also recently been described for the EC cell [73]. Based upon immune-histochemical examinations, there is every reason to postulate that gastric NECs also have their origin in the ECL cell. Gastric NECs are very malignant tumours having a poor prognosis [74]. They may be due to a mutation in a central gene in the regulation of the ECL cell growth. Whether gastrin in normal concentrations has a growth stimulating effect on such mutated cells is unknown.

#### 4.2.2. Gastric Adenocarcinomas

The role of the ECL cell in gastric carcinogenesis has been disputed since the description of ECL cell derived tumours in rodents after long-term profound acid inhibition [8,9]. In reality, this question is connected to the implications of neoplastic tumour cells with endocrine differentiation found in tumours classified as adenocarcinomas [75]. From a biological point of view the distinction between NENs and non-NENs based upon the percentage of tumour cells expressing a NE marker is curious and imprecise since this figure is dependent of the sensitivity of the method applied. By such an attitude neoplasia showing NE positivity in less than 30% of the cancer cells are classified as adenocarcinomas, whereas those displaying positivity in a higher percentage of neoplastic cells are regarded as NENs or mixed neuroendocrine non-neuroendocrine neoplasms (MiNEN), previously known as mixed adeno-neuroendocrine carcinoma (MANEC) [76]. There are many examples of the problems to distinguish between NENs and non-NENs. Thus, Soga et al. reclassified gastric tumours in the African rodent *Mastomys* from adenocarcinomas to NECs [77]. Moreover, Poynter changed the classification of tumours occurring in rodents after long-term profound acid inhibition, from adenocarcinomas [78] to NENs [9]. Similarly, gastric neoplasia in humans have also been reclassified from adenocarcinomas to NENs [79,80]. Moreover, we have shown that many of the tumour cells in signet ring cell carcinomas in humans are NE marker-positive and mucin-negative, indicating that they are NECs [81,82]. When using immunohistochemistry with higher sensitivity, we could show that the carcinoma cells in cancers developed in patients with hypergastrinemia were NE cell marker-positive [83]. NE cell differentiation in signet ring cell carcinomas has also been described by others [84,85]. Long-term hypergastrinemia predisposed to gastric carcinomas in all species examined [38]. In a follow-up, patients with hypergastrinemia at the initial blood sampling developed gastric carcinomas more often than those with normal gastrin values [43]. It has long been known that gastric carcinomas occur almost exclusively in a stomach with gastritis [86]. Since Hp is the dominating cause of gastritis [50], it was therefore natural that Hp early was recognized as the principal factor causing gastric cancer [87]. The mechanism of the carcinogenic effect of Hp has been intensively searched for 25 years. Studies have often focused on changes in chromatin and genes in gastric cells [88,89], but the mechanism has not been found. Since Hp predisposes to gastric cancer only when having induced oxyntic atrophy [90], we have proposed that the carcinogenic effect of Hp infection is mediated by gastrin [91]. 

Histologically, gastric carcinomas often show resemblance to intestinal mucosa/tumours, and the classification systems are as a rule based upon such similarities. They are all classified according to their growth pattern into adenocarcinomas of intestinal type (showing a glandular growth pattern), or diffuse type (showing a non-cohesive growth of the carcinoma cells) according to Laurén [75]. PAS positivity believed to represent mucin is the argument for classifying carcinomas of diffuse types as adenocarcinomas. We have, however, not detected mucin in tumour cells of the diffuse type either by immunohistochemistry or in situ hybridization. Taken into consideration that PAS positivity is not specific for mucin, but only reflecting glycoproteins, we have challenged that these carcinomas are adenocarcinomas. We have focused on the peculiarity in accepting non-specific histochemical methods as the base for tumour classification [92]. In general, gastrin and its target cell, the ECL cell, seem to play a crucial role in gastric carcinogenesis. A proportion of gastric carcinomas express the gastrin receptor even in a late phase [93], which could indicate that these carcinomas may respond to treatment with a gastrin antagonist like netazepide [66]. The prevalence of gastric carcinomas is falling, probably due to a reduction in *Helicobacter pylori* infections. On the other hand, up to 10% of the population uses PPIs [94]. PPIs give profound inhibition of gastric acid secretion resulting in hypoacidity and secondary hypergastrinemia, which, in turn, may predispose to gastric carcinomas [95]. In fact, the first epidemiological studies reporting increased prevalence of gastric carcinomas in PPI users have appeared [96,97]. Figure 1 summarizes the regulation of gastric acid secretion and its connection with gastric neoplasia.

## 5. Conclusion 

Although the prevalence of gastric carcinomas is declining, this disease is still an important cancer, particularly because the prognosis has not improved during the past decades. To improve the outcome, a better understanding of tumorigenesis is mandatory. A better classification based upon biology will probably give rise to new treatment options, including gastrin antagonists. 

## Figures and Tables

**Figure 1 ijms-20-02444-f001:**
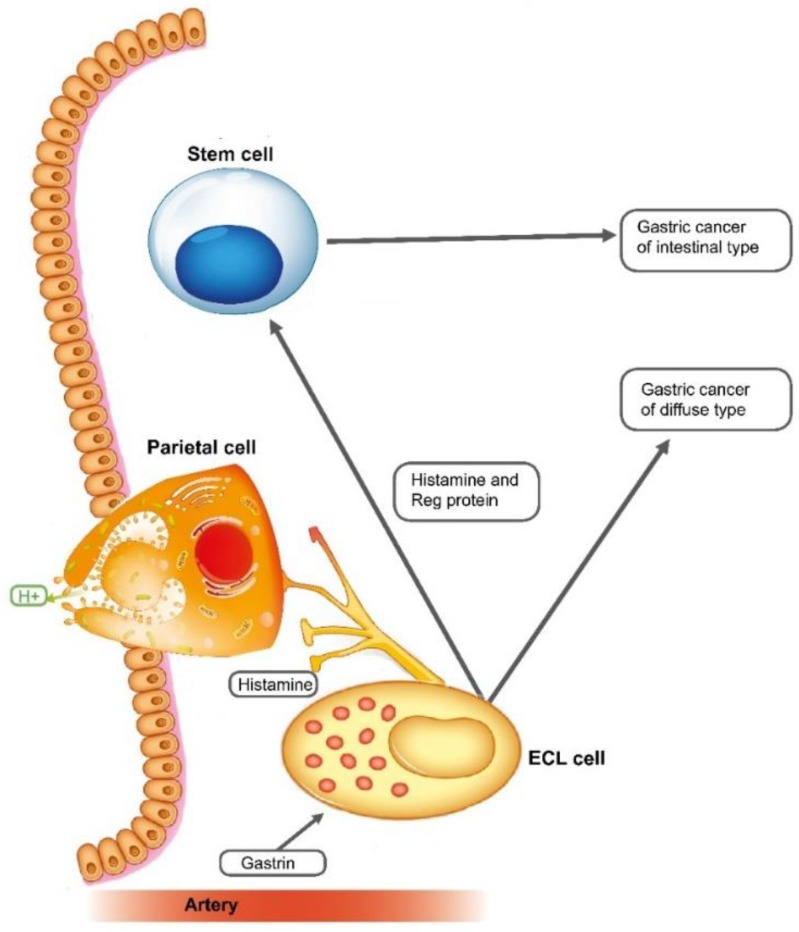
Gastrin and the ECL cell are central in gastric physiology and carcinogenesis.

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
