# Peer review of "The Enterochromaffin-like [ECL] Cell—Central in Gastric Physiology and Pathology"

_ijms, 2019, doi:10.3390/ijms20102444_

Round 1
Reviewer 1 Report
General comments:
This review focuses on the ECL-cell in the stomach. The ECL-cell is a decisive relay-station in the regulation of gastric acid secretion. The initial part of the review is of considerable historical value by giving full credit to Rolf Håkanson and coworkers for their original identification of the ECL-cell, and later for providing solid evidence for the existence of the decisive G (gastrin)-ECL (histamine)-parietal cell (acid) axis in the control of gastric acid secretion.
The authors of the review also describe their own valuable contribution to the understanding of the ECL-cell function, and - not least - to the role of ECL-pathophysiology in diseases of the stomach; both gastric acid diseases and neoplasias.
Specific comments:
The language of the review is occasionally rugged and difficult to follow with too long sentences. It requires some linguistic revision.
Also, a couple of pedagogic illustrations would be helpful for the reader.
Finally, a few more key-references deserve mentioning. For instance the observations from the gastrin-KO and gastrin/CCK2-receptor KO mice that not only is the ECL-cell morphology changed, but also that histamine and acetylcholine have no acid-stimulatory activity without some basal level of gastrin (see 1) Friis-Hansen et al. Impaired gastric acid secretion in gastrin-deficient mice. Am J Physiol 1998;274:G561-G568; and Langhans et al. Abnormal gastric histology and decreased acid production in cholecystokinin-B/gastrin receptor-deficient mice. Gastroenterology 1997;112:280-286.
Author Response
Thank you for the useful comments.
We have hopefully improved the language by splitting long sentences into two or more shorter sentences. Moreover, we have changed some sentences from passive to active mode.
We are very pleased for the suggestions of including the results of mice knockout studies. This has been done, lines 71-79, and included relevant new references, 17-22.
Finally, we have included a figure which we believe that will make the paper easier to understand.
Reviewer 2 Report
In this review, authors have provided evidence from previously published studies that long-term hypergastrinemia results in gastric neoplasia including gastric cancer. Overall the manuscript is well organized and well written. However, I have a few concerns and related suggestions.
The authors did not discuss regarding the the long-term effects of H. pylori-induced epigenetics epigenetic alterations (Chiariotti et al., 2015, Angrisano et al., 2012; Pero et al., 2011) and the dysregulation of defensins (Pero et al., 2017) which may occur during gastric carcinogenesis.
I suggest to briefly include this point or may include a subsection to cover this topic.
Author Response
Thank you for the positive evaluation of our manuscript.
We have added lines 227-231 concerning suggested additions of alternative mechanisms for the carcinogenic effects of Helicobacter pylori, and added two related references, 88-89.